# Eliminating Synaptic Ribbons from Rods and Cones Halves the Releasable Vesicle Pool and Slows Down Replenishment

**DOI:** 10.3390/ijms23126429

**Published:** 2022-06-08

**Authors:** Chris S. Mesnard, Cody L. Barta, Asia L. Sladek, David Zenisek, Wallace B. Thoreson

**Affiliations:** 1Department of Ophthalmology and Visual Sciences, Truhlsen Eye Institute, University of Nebraska Medical Center, Omaha, NE 68198, USA; chris.mesnard@unmc.edu (C.S.M.); cody.barta@unmc.edu (C.L.B.); asia.sladek@unmc.edu (A.L.S.); 2Pharmacology and Experimental Neuroscience, University of Nebraska Medical Center, Omaha, NE 68198, USA; 3Department of Cellular and Molecular Physiology, Yale University, New Haven, CT 06510, USA; david.zenisek@yale.edu

**Keywords:** rods, cones, electroretinogram, exocytosis, mouse, retina, ribbon synapse

## Abstract

Glutamate release from rod and cone photoreceptor cells involves presynaptic ribbons composed largely of the protein RIBEYE. To examine roles of ribbons in rods and cones, we studied mice in which GCamP3 replaced the B-domain of RIBEYE. We discovered that ribbons were absent from rods and cones of both knock-in mice possessing GCamP3 and conditional RIBEYE knockout mice. The mice lacking ribbons showed reduced temporal resolution and contrast sensitivity assessed with optomotor reflexes. ERG recordings showed 50% reduction in scotopic and photopic b-waves. The readily releasable pool (RRP) of vesicles in rods and cones measured using glutamate transporter anion currents (I_A(glu)_) was also halved. We also studied the release from cones by stimulating them optogenetically with ChannelRhodopsin2 (ChR2) while recording postsynaptic currents in horizontal cells. Recovery of the release from paired pulse depression was twofold slower in the rods and cones lacking ribbons. The release from rods at −40 mV in darkness involves regularly spaced multivesicular fusion events. While the regular pattern of release remained in the rods lacking ribbons, the number of vesicles comprising each multivesicular event was halved. Our results support conclusions that synaptic ribbons in rods and cones expand the RRP, speed up vesicle replenishment, and augment some forms of multivesicular release. Slower replenishment and a smaller RRP in photoreceptors lacking ribbons may contribute to diminished temporal frequency responses and weaker contrast sensitivity.

## 1. Introduction

Ribbons are electron-dense structures in sensory neurons that support sustained exocytosis, allowing synapses to convey sensory information [1,2]. The core of each ribbon is constructed around a transcript variant of CtBP2 known as RIBEYE [3]. RIBEYE is comprised of a structural A-domain and an enzymatic B-domain that is nearly identical to the more ubiquitous short form of CtBP2. Ribbons serve as molecular scaffolds for synaptic vesicles, supporting the recruitment of new vesicles to docking sites on the ribbon and delivering them to the active zone [3,4,5,6,7]. The release from cones occurs exclusively at ribbon release sites, whereas rods are capable of both ribbon and non-ribbon release [2]. Synaptic vesicles at the base of each ribbon form the RRP, and fusion of these vesicles provides space for vesicles further up the ribbon to descend toward the membrane to replenish this pool [8,9,10]. The RRP provides a fast transient component of release and is comprised of the number of vesicles contacting the plasma membrane along the bottom one or two rows of the ribbon [11,12,13,14]. The reserve pool (RP) consists of the vesicles attached further up the ribbon that can replenish the RRP. The RRP and RP are primed and ready for release, meaning they have undergone all ATP-dependent steps required for fusion [15,16,17]. A third cytoplasmic reserve pool replenishes synaptic ribbons, allowing release to be maintained indefinitely [11,18]. In addition to tethering vesicles above the active zone, ribbons are thought to play a number of other roles including priming vesicles for release [16,17], regulating the rate of replenishment [7,19,20,21,22,23,24], and facilitating multivesicular release [25,26,27,28].

Previous work on synaptic ribbons showed that deletion of RIBEYE from mouse rods eliminated ribbons and reduced the number of membrane-associated vesicles by 60% [29]. Measurements of exocytotic capacitance changes in rods suggested a 75% reduction in the physiological RRP after RIBEYE deletion [30]. In rod bipolar cells, fast and sustained release were both greatly reduced while release kinetics remained unchanged in the absence of ribbons [29]. Electroretinograms (ERGs) showed a reduced b-wave under scotopic conditions, consistent with reduced release from rods [31,32]. However, photopic b-waves were unchanged, suggesting release from cones was not affected by RIBEYE deletion [31]. There were surprisingly modest deficits in responses of ON alpha retinal ganglion cells involving reduced responses to high-frequency flicker and diminished contrast sensitivity [33].

To further investigate the roles played by retinal ribbons, we studied the visual function using optomotor response (OMR) assays, electroretinograms (ERGs), and whole-cell recordings in mice lacking ribbons. We studied the release from individual rods and cones using presynaptic glutamate transporter currents as an assay for glutamate release. We also studied cone release by recording postsynaptic currents evoked in horizontal cells by optogenetic stimulation of cones. In *Ribeye* KI mice, the enzymatic B-domain was replaced with GCamP3 and flanked with loxP sites for Cre-dependent deletion. As reported recently [34], we discovered that substituting GCamP3 for the RIBEYE B-domain eliminated ribbons throughout the retina. From OMR assays, we found that eliminating retinal ribbons reduced temporal resolution and contrast sensitivity. Loss of ribbons from rods and cones also halved scotopic and photopic ERG b-waves, halved the RRP, and slowed down vesicle replenishment. Loss of ribbons also shortened multiquantal release events that may help encode visual information at visual threshold [35,36]. Our results suggest that by expanding the RRP and speeding up replenishment, rod and cone ribbons can play an important role in improving visual performance.

## 2. Results

### 2.1. Anatomical Characterization

To study the roles played by ribbons in rods and cones, we began with *Ribeye* KI mice (B6;129S6-Ctbp2tm1Sud/J) that express a modified form of RIBEYE in which the enzymatic B-domain was replaced with GCamP3 followed by a stop codon, with the sequence flanked by loxP sites. This knock-in strategy allows uninterrupted expression of the short isoform of CtBP2, but replaces the B-domain from RIBEYE independent of Cre expression, while the Cre recombinase is expected to remove the A-domain [29]. We crossed these with Rho-iCre or HRGP-Cre mice expressing the Cre recombinase in rods or cones to eliminate the RIBEYE A-domain selectively from these photoreceptor cells. We examined retinas of control, conditional *Ribeye* knockouts, and the knock-in mice in which GCamP3 replaced the B-domain.

In a control C57Bl6J mouse retina, antibodies to the B-domain of CtBP2 yielded punctate labeling of photoreceptor ribbons in the OPL and bipolar cell ribbons in the IPL (Figure 1A). The B-domain antibody labels both RIBEYE and the short form of CtBP2 expressed in the inner nuclear layer (INL). For the retinal sections in Figure 1, we used a rhodamine-conjugated secondary antibody to visualize immunolabeling of RIBEYE B-domain. We co-labeled these retinas with FITC-conjugated PNA to stain the bottoms of cone pedicles. FITC-conjugated PNA also labels cone inner segments and weakly labels the IPL. Labeling of the OPL by the B-domain antibody was eliminated in conditional knockout (CKO) mice where RIBEYE should be eliminated from both rods and cones (*Ribeye*^fl/fl^ × HRGP-Cre × Rho-iCre; Figure 1B).

In principle, *Ribeye* KI mice should express a protein consisting of the A-domain followed by GCamP3. Because the B-domain was replaced by GCamP3, labeling with the B-domain antibody was also eliminated from bipolar cell terminals in the IPL of the rod/cone *Ribeye* CKO mice (Figure 1B). Similarly, replacing the B-domain with GCamP3 in the *Ribeye* KI mice eliminated labeling by the RIBEYE B-domain antibody of photoreceptor terminals in the OPL and bipolar cell terminals in the IPL (Figure 1C). The rod/cone *Ribeye* CKO and *Ribeye* knock-in mice both showed staining of cell bodies in the INL due to expression of the short form of CtBP2.

In the control C57Bl6J mice, labeling with antibodies to the RIBEYE-specific A-domain produced strong labeling of rod and cone terminals in the OPL along with weaker labeling of bipolar cell ribbon synapses in the IPL (Figure 2A). To visualize these sections, we used an FITC-conjugated secondary antibody to label RIBEYE along with rhodamine-conjugated PNA to label cone pedicles. When we examined the *Ribeye* KI mice, labeling for the RIBEYE A-domain was entirely absent from the OPL and IPL (Figure 2B). This suggests that replacing the B-domain with GCamP3 disrupts RIBEYE stability, expression, and/or trafficking to the synapse. The A-domain antibody labels only RIBEYE, not the shorter form of CtBP2, so this antibody did not label the INL.

As a further test for the presence or absence of ribbons in these mice, we used transmission electron microscopy. Figure 3 shows examples of synaptic ribbons (arrows) from rods and cones in the control C57Bl6J, rod*^Ribeye^*^CKO^, cone^*Ribeye*CKO^, and *Ribeye* KI mice. The top row shows rod terminals; the bottom row shows cone terminals. Prominent electron-dense ribbons were evident in rods (Figure 3A) and cones (Figure 3B) from the C57Bl6 mice. As expected from Cre-dependent deletion of *Ribeye*, ribbons were absent from rods of the rod*^Ribeye^*^CKO^ mice (Figure 3C) and from cones of the cone^*Ribeye*CKO^ mice (Figure 3D). However, ribbons were also absent from rods (Figure 3E) and cones (Figure 3F) of the *Ribeye* KI mice where the B-domain was replaced with GCaMP3. Presynaptic arciform densities (arrows), trough-like structures that normally lie beneath photoreceptor ribbons, remained intact. These data support immunohistochemical results that both the KI and the conditional KO mice lacked ribbons in both rods and cones, and support previous findings of Shankhwar et al., 2022.

### 2.2. Optomotor Responses

We assessed visual behavior in the mice lacking retinal ribbons from their optomotor responses to rotating gratings (Figure 4). Visual acuity was defined as the maximum spatial frequency evoking an optomotor response. *T*-tests corrected for multiple comparisons (false discovery rate approach) on visual acuity estimates showed a significantly higher acuity in control mice (*n* = 12) relative to the *Ribeye* KI mice (*n* = 14) at 0.3 Hz (*p* = 0.00116). Tracking was maintained for up to 0.3 Hz in the control mice but almost completely abolished by that frequency in the *Ribeye* KI mice.

Contrast sensitivity was defined as the minimum stimulus contrast evoking an optomotor response. *T*-tests corrected for multiple comparisons revealed that the control mice were significantly more sensitive than the *Ribeye* KI mice to contrasts of 0.04–0.64%. These data suggest that despite modest deficits in ON alpha ganglion cells [33], ribbons can play an important role in shaping visual performance.

### 2.3. Electroretinograms

To assess the role of ribbons in synaptic transmission from rods and cones, we performed ERGs on mice in vivo. Figure 5A shows representative waveforms evoked by high-intensity (−3 dB) 20 ms flashes applied to the dark-adapted mice under scotopic conditions. Scotopic b-wave amplitudes were significantly reduced (*p* = 0.001, one-way ANOVA) in the *Ribeye* KI (*p* = 0.0045, Holm–Sidak’s multiple-comparisons test), rod*^Ribeye^*^CKO^ (*p* = 0.0022), and cone*^Ribeye^*^CKO^ (*p* = 0.0034) mice compared to the C57Bl6J controls (Figure 5B). The b-waves were reduced to the same extent in all three mouse lines, with a 50% reduction at brighter flash intensities. The a-waves were slightly but not significantly smaller in the *Ribeye* KI (*p* = 0.25), rod^*Ribeye*CKO^ (*p* = 0.33), and cone*^Ribeye^*^CKO^ (*p* = 0.41) mice compared to the C57Bl76J control mice (Figure 5C). This is consistent with impaired synaptic transmission from photoreceptors to bipolar cells. B-wave latencies were also not changed (Figure 5D).

Figure 6A shows example b-waves obtained under photopic conditions. Like scotopic responses, photopic b-wave amplitudes were also reduced significantly (*p* = 0.0025, ANOVA) compared to the C57Bl6J mice in the *Ribeye* KI (*p* = 0.02), rod*^Ribeye^*^CKO^ (*p* = 0.01), and cone*^Ribeye^*^CKO^ (*p* = 0.009) mice with a 50% reduction at the highest flash intensities (Figure 6B). A-wave amplitudes and b-wave latencies showed no differences between strains (Figure 6C,D). The diminished scotopic b-wave was consistent with prior studies on whole animal KO mice and suggests impaired synaptic transmission from rods [31,32]. Our finding of diminished photopic b-waves differs from results of a previous study by Fairless et al., 2020, and suggests loss of ribbons also impairs transmission from cones [31].

### 2.4. Whole-Cell Recordings

To study the release in individual rods and cones, we recorded anion currents activated by glutamate binding to plasma membrane glutamate transporters. EAAT2 and EAAT5 transporters in rods and cones exhibit large anion currents upon glutamate binding and thus provide a presynaptic measure of glutamate release [37]. To enhance glutamate transporter anion currents (I_A(glu)_), we used a highly permeant anion, thiocyanate (SCN^–^), as the principal anion in our patch pipette solution. Step depolarization applied to cones (Figure 7) or rods (Figure 8) stimulates inward currents during the step that reflects the summed activity of both calcium currents (I_Ca_) and I_A(glu)_. Upon termination of the depolarizing step, I_Ca_ declines rapidly with a time constant of <2 ms [32]. The remaining inward current reflects the activity of I_A(glu)_ and declined more slowly with a time constant of ~50 ms. To assess the glutamate release, we measured I_A(glu)_ charge transfer beginning 2 ms after step termination. Long depolarizing steps sometimes triggered Ca^2+^-activated Cl^−^ currents that continued well beyond the 2 s acquisition period [36]. We subtracted contributions from these maintained tail currents by measuring I_A(glu)_ charge transfer relative to the baseline current at the end of the 2 s acquisition period. We varied the step duration to examine different phases of release.

Figure 7 shows examples of I_A(glu)_ evoked by steps of 50 and 200 ms recorded from the same cone in a control C57Bl6J retina (black traces). Passive membrane properties were subtracted using P/6 protocols. Overlaid on these control responses are recordings from a cone in a cone*^Ribeye^*^CKO^ retina (red traces). The absence of ribbons in the cone*^Ribeye^*^CKO^ mice led to smaller I_A(glu)_ over the entire range of step durations. The I_A(glu)_ charge transfer rose with a similarly rapid initial time constant in the control (3.3 ms) and cone*^Ribeye^*^CKO^ cones (2.4 ms; exponential fit constrained to the first 25 ms; F-test, *p* = 0.83). I_A(glu)_ events arising from the fusion of individual vesicles decline with a time constant of ~50 ms [36]. This rate is primarily determined by the rate at which glutamate molecules bound to the transporter are retrieved [37]. Thus, I_A(glu)_ measurements provide a snapshot of release that occurred over the previous 50 ms or so. Retrieval of the glutamate released at the beginning of the step may thus explain a small decline in I_A(glu)_ measured after steps of 50–100 ms (Figure 7 and Figure 8). Differences were significantly different at timepoints of 5, 25, and 100 ms (*p*-values adjusted for multiple comparisons, Holm–Sidak method, *p* = 0.029, 0.018, and 0.035, respectively). These data show that the initial release kinetics are similar, but loss of ribbons reduces the RRP evoked in cones by brief depolarizing steps to roughly half the size found in control cones.

Figure 8 shows example recordings from a rod in a C57Bl6J control retina (black traces) and a rod from rod*^Ribeye^*^CKO^ retina (red traces) evoked by depolarizing steps of 10 and 200 ms. As with cones, the absence of ribbons in rod*^Ribeye^*^CKO^ mice reduced I_A(glu)_ at all step durations. I_A(glu)_ charge transfer rose quickly in both control (4.5 ms) and rod*^Ribeye^*^CKO^ rods (1.4 ms; *p* = 0.44) and, like cones, declined slightly when measured after steps of 50–100 ms (Figure 8). Differences between rods with and without ribbons were significant at 25, 50, and 100 ms (adjusted *p*-values, Holm-Sidak method for multiple comparisons, *p* = 0.028, 0.048, and 0.042, respectively). Like cones, the absence of ribbons halved the fast component of I_A(glu)_ in rods compared to control rods. A reduction in the size of the RRP is consistent with ultrastructural measurements of the number of membrane-associated vesicles and capacitance measurements of the RRP made using capacitance techniques in *Ribeye* KO rods [29,32].

Inward currents during depolarizing steps reflect a combination of I_Ca_ and I_A(glu)_. When we measured I_Ca_ by itself using a ramp voltage protocol (0.5 mV/ms, from −90 to +60 mV), we did not see any significant change in the peak amplitude of I_Ca_ in rods or cones (control rods: 8.4 ± 4.85 pA, *n* = 26; rod*^Ribeye^*^CKO^: 8.8 ± 5.04 pA, *n* = 20; control cones: 72.6 ± 55.5 pA, *n* = 33; cone*^Ribeye^*^CKO^: 59.7 ± 41.4 pA, *n* = 26). This suggests that the reduction in release from the rods and cones lacking ribbons was not due to a reduction in I_Ca_.

When rods are voltage-clamped at their typical resting potential in darkness of −40 mV, we observed bursts of multiquantal release that occurred at semiregular intervals [35,36]. The size and regularity of these multiquantal release events may help rod bipolar cells discriminate single photon responses during an ongoing release from rods [35]. As illustrated in Figure 9, regular and periodic multiquantal release events remained in the rods lacking ribbons, but the duration of each multiquantal event was halved in the rod*^Ribeye^*^CKO^ mice (Figure 9; *p* = 0.003, nested *t*-test, *n* = 10 control rods, 167 events; *n* = 5 rod*^Ribeye^*^CKO^ rods, 98 events). Because of the rapid rundown of release in mouse rods, we did not obtain a sufficiently large sample to accurately analyze inter-burst intervals. Combined with the evidence that loss of ribbons halves the RRP, these data support the hypothesis that each multiquantal event empties the available RRP and that the ribbon plays an important role in defining the size of these bursts [36].

Miniature I_A(glu)_ events arising from individual vesicle fusion events did not differ between control and *Ribeye* CKO rods held at −70 mV in mean amplitude (control: 6.1 ± 2.06 pA, *n* = 5 rods; rod*^Ribeye^*^CKO^: 4.7 ± 1.06 pA, *n* = 6 rods) or half-width (control: 31.7 ± 6.68 ms; rod*^Ribeye^*^CKO^: 32.8 ± 2.93 ms; Figure 9D). Interevent intervals measured in 30 s trials also did not differ between the control rods (1767 ± 537.6 ms, *n* = 8 rods) and the rods lacking ribbons (1911 ± 710.7 ms, *n* = 7 rods; *p* = 0.54, nested *t*-test). Miniature I_A(glu)_ events recorded at −70 mV did not differ between the control and cone*^Ribeye^*^CKO^ cones at the amplitude (control: 9.2 + 3.85 pA, *n* = 10 cones; cone*^Ribeye^*^CKO^: 9.2 + 3.33 pA, *n* = 8 cones) or in half-width (control: 32.2 ± 7.3 ms; cone*^Ribeye^*^CKO^: 28.3 ± 8.4 ms; Figure 9D).

An important function ascribed to ribbons is the ability to regulate delivery and replenishment of vesicles to release sites at the base of the ribbon [7,19,20,21,22,23,24]. However, hair cells and bipolar cells in animals lacking RIBEYE did not show differences in release kinetics when studied with long depolarizing steps or trains of stimuli [29,38,39,40]. To test the role of ribbons in replenishment, we used a paired pulse protocol. Figure 10 shows examples of I_A(glu)_ recorded from a control C57Bl6J rod and a *Ribeye* KI rod that lacked ribbons. The 25 ms test steps in these examples were separated by 500 ms. As illustrated by these responses and summarized in Figure 10C, recovery was slower in the absence of ribbons. We fit recovery of the I_A(glu)_ charge transfer from paired pulse depression with a single exponential function. In the absence of ribbons, the time constant doubled (*p* = 0.026; extra sum-of-squares F-test) from 183 ms in the C56Bl6 rods (*n* = 10 rods; rate constant, K = 0.00547 ± 0.00102 ms, SEM) to 422 ms in the *Ribeye* KI rods (*n* = 12; K = 0.002372 ± 0.000351 ms). Individual values differed significantly at the 500 ms interval (*p* = 0.002, *t*-tests corrected for multiple comparisons).

To test vesicle replenishment at cone synapses, we employed a different approach using optogenetic stimulation of cones. For these experiments, we prepared horizontal slices of a retina to expose horizontal cells while retaining intact connections with the cones below. To activate cones optogenetically, we crossed the cone*^Ribeye^*^CKO^ mice with Ai32 mice that express ChRh2 under control of the Cre recombinase. Figure 11A shows a horizontal cell filled with sulforhodamine B through a patch pipette. In this example, the retina was also loaded with a Ca^2+^-sensitive dye, Cal520, which preferentially labels cones due to the large membrane surface area of their outer segments. As illustrated in Figure 11, stimulating cones with a brief LED flash (1 ms) evoked fast inward currents that averaged 682 ± 221 pA in HCs from the control C57Bl6 retinas (*n* = 11 HCs in four mice) and 712 ± 229 pA in HCs from the cone*^Ribeye^*^CKO^ mice (*n* = 10 in four mice). The amplitude/intensity relationships fit with a logistic function did not differ between the control and cone *Ribeye* CKO retinas (F-test, *p* = 0.99). In addition to rapid inward optogenetic currents, LED light flashes produce weak cone responses yielding slower outward currents in horizontal cells. In paired pulse depression experiments, we measured the response amplitude relative to the outward current immediately attained after each flash, assuming this to be the condition with the least amount of glutamate release. For these experiments, we blocked AMPA receptor desensitization by including cyclothiazide (CTZ; 0.1 mM) in the bath.

Using pairs of LED flashes separated by varying intervals, optogenetically evoked responses recovered with a time constant of 214 ms (K = 0.004665 ± 0.0003175) in the control retinas but more slowly in the cone*^Ribeye^*^CKO^ retinas with a time constant of 366 ms (K = 0.002732 ± 0.0003716 ms; *p* = 0.0002, F-test). These differences are illustrated by the example recordings in Figure 11 and the graph in Figure 11D. Figure 11B shows a series of paired pulse trials overlaid on one another from a control mouse retina while Figure 11C shows paired pulse trials from a cone*^Ribeye^*^CKO^ horizontal cell. When CTZ was omitted, the presence of glutamate receptor desensitization produced similarly slow recovery in both the control and cone*^Ribeye^*^CKO^ mice (control: 361 ms, rate constant = 0.00277 ± 0.00036 ms, *n* = 12; cone*^Ribeye^*^CKO^: 318 ms, rate constant = 0.00314 ± 0.00041 ms, *n* = 12). These two different experimental approaches in rods and cones both show that eliminating ribbons halves replenishment rates in photoreceptors.

In another approach to testing replenishment, we applied a 3 s train of 1 ms optogenetic pulses each separated by 100 ms (Figure 12). As above, we conducted these experiments in the presence of CTZ (0.1 mM) to block AMPA receptor desensitization in horizontal cells. Figure 12A shows the average optogenetically evoked responses from horizontal cells in the control (black) and cone*^Ribeye^*^CKO^ (red) retinas. In both, the first LED flash evoked a large, fast inward current due to the optogenetically evoked release of glutamate from cones. This fast inward current was followed by a slower outward current reflecting the decline in glutamate release from cones following light-evoked hyperpolarization due to activation of endogenous opsins. The second optogenetically evoked inward current in the train was much smaller than the initial response and the responses recovered. Responses recovered more rapidly (τ = 221 ms) and to a greater extent in the C57Bl6 retinas than in horizontal cells from the cone*^Ribeye^*^CKO^ retinas (τ = 377 ms). The rates of recovery in the pulse train were similar to the rates of recovery obtained from measurements of paired pulse depression. Faster replenishment in cones with intact ribbons is also evident from the steeper slope of the cumulative charge transfer compared to the charge transfer in horizontal currents recorded in the cone*^Ribeye^*^CKO^ retinas (Figure 12B; dashed lines show 95% confidence intervals). 

## 3. Discussion

We examined the roles played by ribbons in rods and cones by studying conditional knockout mice in which *Ribeye* was selectively eliminated from rods or cones as well as *Ribeye* KI mice with a GCamP3-tagged version of RIBEYE that also lacked ribbons. The absence of ribbons in *Ribeye* KI mice was also recently reported by Shankhwar et al. (2022).

From optomotor responses, we found that contrast sensitivity was halved in the *Ribeye* KI mice lacking ribbons. This was a more profound deficit than seen in ON alpha retinal ganglion cells [33], perhaps because optomotor responses are driven by inputs from a different population of retinal ganglion cells consisting of ON/OFF directionally selective ganglion cells [41].

The initial increase in the I_A(glu)_ charge transfer as a function of test step duration represents release of the RRP of vesicles. We saw similarly fast release kinetics with or without ribbons but a 50% reduction in the number of vesicles released in the first 25 ms when ribbons were absent. When stimulated with longer steps, the amount of release in rods or cones lacking ribbons remained roughly half that of the control retinas. Previous ultrastructural studies concluded that eliminating ribbons reduced the density of vesicles at synaptic junctions by ~41% with a ~60% overall reduction in the number of docked vesicles [29]. Using capacitance measurements of release from rods in *Ribeye* KO mice, Grabner and Moser found a 75% reduction in the RRP [32]. In amphibian and mammalian retinas, the physiological RRP is equal in size to the number of vesicles tethered along the bottom two rows of the ribbon in contact with the adjacent plasma membrane [2,42]. One interpretation of our results is that without a ribbon, only a single row of vesicles can dock to the membrane, reducing the RRP by half. Reducing the size of the RRP reduces the number of vesicles available to encode the range of luminance. By coarsening resolution, this reduces sensitivity to fine changes in contrast [43]. These effects may contribute to deficits in contrast sensitivity seen with OMR (present study) and in ganglion cells from retinas lacking ribbons [33].

Loss of ribbons can disrupt the clustering of Ca^2+^ channels [29,39]. With very brief steps of 1 ms, Grabner and Moser (2021) saw a small difference in I_Ca_ between control and *Ribeye* KO rods, but they found that this difference disappeared with longer steps. Consistently with their results using longer test steps, we saw no significant difference in the rod I_Ca_ amplitude measured with ramp protocols between the mice with and without ribbons. These data suggest that the reduction in the RRP seen with strong test steps is not due to reduced Ca^2+^ influx, but rather due to a diminished number of releasable vesicles.

Consistent with a 50% reduction in the RRP, we found that scotopic and photopic b-waves, which reflect ON bipolar cell activity, were both halved without ribbons. The a-wave, which reflects photoreceptor light responses, did not show a significant reduction. The reduced scotopic b-wave agrees with earlier studies [31,32]; however, a previous study on *Ribeye* KO mice found no reduction in the photopic b-wave [31]. The reason for this difference remains unclear.

In addition to shaping the RRP, an important role ascribed to ribbons is regulation of vesicle replenishment. Vesicles have been found to replenish rod and cone synapses in both mammalian and salamander retina with time constants of a few hundred milliseconds [22,44,45]. It has been proposed that ribbons may speed up replenishment, but also that ribbons may slow down replenishment [43,46]. Our results from I_A(glu)_ measurements in rods and optogenetic stimulation of cones showed that the rate of replenishment slowed by ~50% in the absence of ribbons, supporting the proposal that ribbons, at least in photoreceptors, speed up replenishment. Eliminating ribbons from rod bipolar cells reduced sustained release, perhaps by slowing down replenishment [29]. In studies on inner hair cells, capacitance measurements of release showed no difference in recovery from paired pulse depression with and without ribbons [38,40], but analysis of downstream spiral ganglion neurons nevertheless suggested impaired replenishment [40] The role of ribbons in replenishment in hair cells and bipolar cells remains to be fully resolved.

In ON alpha retinal ganglion cells in *Ribeye* KO mice, half-maximal responses to flickering stimuli fell from ~4 to ~2 Hz under mesopic conditions [33]. We saw diminished optomotor responses to 0.3 c/deg spatial frequency grating rotating at 12 deg/s under mesopic conditions, consistent with diminished responses to 4 Hz stimuli. In the absence of further assumptions, a twofold slowdown of replenishment in rods and cones from time constants of ~200 to ~400 ms predicts a twofold slowing in frequency responses from ~5 Hz to ~2.5 Hz. However, other factors shape ganglion cell response kinetics, including the rate of vesicle replenishment in bipolar cells and properties of glutamate receptor desensitization. At OFF bipolar cell synapses in ground squirrel retinas, the kinetics of receptor desensitization and recovery can actually speed up recovery [44]. In our experiments, when AMPA receptor desensitization remained intact at the cone-to-horizontal-cell synapses, recovery from paired pulse depression at cone synapses of the control retinas slowed to the rates seen in the cones lacking ribbons.

Another postulated role for ribbons is that they may facilitate synchronous multivesicular release [25,27,28,47]. We did not see significant differences in the amplitude distributions of synchronous release events in rods or cones with or without ribbons, but the number of large synchronous events was too small to exclude the possibility that they were impaired by loss of ribbons. We also found no significant difference in the frequency of individual spontaneous release events in rods, supporting evidence from TIRFM experiments in amphibian rods that spontaneous release occurs primarily at non-ribbon release sites [48].

At the typical rod membrane potential in darkness of −40 mV, rods exhibit large bursts of multivesicular release that occur at regular intervals [36]. These release events are not tightly synchronized with one another, but instead involve the sequential release of multiple vesicles with each event thought to empty the RRP [36]. The size and regularity of these release events can improve the ability of rod bipolar cells to detect and distinguish changes in release produced by small single photon voltage responses in rods [35]. It has been proposed that descent down the ribbon could provide a mechanism for regularizing release [49]. However, we found that these large multiquantal release events retained a regular pattern of release in the rods lacking ribbons, but the duration of each multiquantal release event was halved, supporting the hypothesis that they empty the available RRP.

In summary, our results show that loss of ribbons halves the RRP and halves the rate of vesicle replenishment in both rods and cones. Reducing the size of the releasable pool of vesicles can limit the ability to encode fine contrast steps, and slowing down replenishment can limit temporal resolution. Although other mechanisms are involved, these effects may contribute to diminished contrast sensitivity and slower temporal frequency responses of retinal ganglion cells [33] and the visual function assessed using optomotor responses. Modeling studies of bipolar cell ribbon synapses suggested that vesicles can be delivered rapidly to release sites even without ribbons, allowing high rates of sustained release [8]. This may help to explain why, despite the prominence of ribbons adorned with vesicles poised for release, eliminating ribbons does not have more catastrophic effects on the release from rods, cones, and bipolar cells.

## 4. Materials and Methods

### 4.1. Mice

Details of HRGP-Cre, Rho-iCre, and *Ribeye* KI mice have been described previously [29,50,51], and these mice are all available from Jackson Laboratories (Bar Harbor, ME, USA) (HRGP-Cre: Tg(OPN1LW-cre)4Yzl/J, Ribeye: B6;129S6-Ctbp2tm1Sud/J, and Rho-iCre: B6.Cg-Pde6b+Tg(Rho-iCre)1Ck/Boc; RRID: 015850). Rho-iCre and *Ribeye* KI mice were created on the C57Bl6J background. HRGP-Cre mice were created on the background of FVB mice and back-crossed into C56Bl6J mice. Rho-iCre and HRGP-Cre mice selectively express the Cre recombinase in rods and cones, respectively. To create mice lacking RIBEYE in rods and cones (rod*^Ribeye^*^CKO^ and cone*^Ribeye^*^CKO^ mice), we crossed *Ribeye* KI mice with Rho-iCre and HRGP-Cre mice. When targeting cones for whole-cell recording, we also crossed HRGP-Cre mice with a Cre reporter line expressing td-Tomato (Jackson Laboratories, B6.Cg-Gt(ROSA)26Sortm14(CAG-tdTomato)Hze/J). We used mice of both sexes aged between 4–8 weeks.

In accordance with the AVMA Guidelines for the Euthanasia of Animals, euthanasia was performed by CO_2_ asphyxiation followed by cervical dislocation. All animal care and handling protocols were approved by the University of Nebraska Medical Center Institutional Animal Care and Use Committee.

### 4.2. Electroretinography

Electroretinograms (ERGs) were recorded from mice in vivo using a UTAS Sunburst ganzfeld illuminator (LKC, Gaithersburg, MD, USA, LKC-UTAS-SB). The mice were dark-adapted for ~12 h before experiments and then anaesthetized via intraperitoneal injection with a mixture of ketamine and xylazine (100 mg/kg ketamine, 10 mg/kg xylazine). The core temperature of the mice was maintained at 37 °C using a heating pad. Tropicamide and proparacaine ophthalmic solution (0.5%) were administered topically to the left eye before the mouse was secured to a platform. A gold wire ring recording electrode was centered on the left cornea. Subcutaneous ground and reference electrodes were placed at the base of the tail and under the scalp, respectively.

Recordings performed under dark-adapted (scotopic) conditions used a series of flashes of increasing intensity: −51 dB, −45 dB, −39 dB, −33 dB, −27 dB, −21 dB, −15 dB, −9 dB, −3 dB, and +5 dB. Ten flashes were presented at each intensity, separated by 10 s for steps 1–9 and 20 s between flashes at the highest intensity. Light-adapted (photopic) protocols were performed following background adaptation for 10 min with green light (40 cd/m^2^) and conducted with the same background using six intensity steps: −6, −3, 0, 4, 7, and 13 dB, each with 25 sweeps separated by 3 s apiece. ERG a-waves provide a measure of photoreceptor responses and were measured from baseline to the bottom of the initial inward-going negative potential. ERG b-waves reflect the responses of ON bipolar cells and were measured from the trough of the a-wave to the peak of the positive-going b-wave.

### 4.3. Optomotor Assay

#### 4.3.1. Chamber Design

The visual function assay chamber was fabricated with interior dimensions of 54 × 54 × 32 cm (W × D × H) and a floor covered with mirrored glass. A circular platform (diameter = 5 cm) was elevated 16 cm above the chamber floor. Four LCD monitors (Dell P2419H; 60 Hz refresh rate) were fit into slots fabricated around the chamber. Each monitor subtended a 90° × 58.1° visual angle with a pixel resolution of 18.5 pixels per degree with respect to the platform center. A camera (Allied Vision Mako G-158B; 60 Hz frame rate) was mounted 17.5 cm above the circular platform, providing a 71.1° × 54.4° field of view with a pixel resolution of 20.4 pixels per degree. The camera and the displays were controlled by MATLAB (R2019a; MathWorks, Natick, MA, USA) installed on a Windows PC (Dell OptiPlex 5060; Intel Core i7-8700 CPU @ 3.2 GHz; 32 GB RAM; NVIDIA Quadro M2000 GPU).

#### 4.3.2. Stimulus Design

Visual stimuli were generated and presented using Visual Psychophysics Toolbox. The stimuli were comprised of vertical square wave gratings with spatial frequency manipulated to induce the illusion of a virtual cylinder with identical spatial frequency across the entire display. Stimulus velocity was fixed to 12 degrees/second across all study procedures. Visual acuity was assessed by presenting 100% contrast gratings at nine spatial frequencies (cycles/degree): 0.025, 0.05, 0.10, 0.20, 0.30, 0.35, 0.375, 0.40, and 0.45. Contrast sensitivity was assessed by presenting 0.20 cycles/degree gratings with the following luminance contrasts (%): 0.01, 0.02, 0.04, 0.08, 0.16, 0.32, 0.64, and 0.96. Luminance was measured using a luminance meter (Konica Minolta LS-150).

#### 4.3.3. Procedures

All the animals were tested between 8:00 AM and 1:00 PM in a room with lights off and ambient illumination of 2.4 cd/m^2^. No more than eight animals were tested within a single day. Prior to stimulus presentation, the animals were placed on the elevated circular platform and allowed to acclimate for at least 5 min. Acclimation periods were terminated when the gross movements in the body positioned were minimized. Stimulus presentation procedures for each trial were as follows. Each trial began with a 2 s blank display (5.7 cd/m^2^). Next, a static square wave grating of predetermined spatial frequency and contrast was presented for 0.33 s. Following the static display, the grating began moving to the left or right; initial stimulus direction was randomly determined. The stimulus motion lasted for 10 s and the stimulus direction reversed halfway through the stimulus motion period. A blank intertrial interval of 30 s followed each stimulus motion sequence. There were 9 and 16 total trials in the visual acuity and contrast sensitivity protocols, respectively. Testing for each mouse took no longer than 30 min to complete.

The videos underwent manual and automated processing. Manual processing routines were performed using custom MATLAB software and involved trained personnel counting the number of frames during which the animals’ head movements appeared slow and reflexive—as opposed to more rapid and volitional—and moved in the same direction as the stimulus. Tracking performance was calculated as the percentage of counted frames relative to the total stimulus movement frames.

### 4.4. Whole-Cell Recordings

Whole-cell recordings from rods and cones were performed using flatmount preparations of isolated retina. An eye was enucleated after euthanizing the mouse and placed in Ames’ medium (US Biological, Salem, MA, USA; RRID:SCR_013653) bubbled with 95% O_2_/5% CO_2_. The cornea was punctured with a scalpel and the anterior segment was removed. The retina was isolated by cutting optic nerve attachments. After making 3–4 fine cuts at the opposite poles, the retina was flattened onto a glass slide in the perfusion chamber with the photoreceptors facing up. The retina was anchored in place with a brain slice harp (Warner Instruments, cat. No. 64-0250). The perfusion chamber was placed on an upright fixed-stage microscope (Nikon E600FN) with a 60× water-immersion long-working-distance objective (1.0 NA). Flatmount preparations were superfused with room temperature Ames’ solution bubbled with 95%/5%CO^2^ at ~1 mL/min.

Patch-recording electrodes were pulled on a Narishige (Amityville, NY, USA) PP-830 vertical puller using borosilicate glass pipettes (1.2 mm outer diameter, 0.9 inner diameter, with internal filament; World Precision Instruments, Sarasota, FL, USA). The pipettes had tip diameters of 1–2 μm and resistances of 10–15 MΩ. To clear away outer segments and expose inner segments of rods and cones, we used two techniques. In some experiments, we applied suction through a large-bore patch pipette to gently vacuum up outer segments. In others, we used a “tissue print” method where we gently pressed a piece of nitrocellulose filter paper onto the retina for a few seconds and then removed it to pull away adherent outer segments. Rod inner segments were targeted with positive pressure using recording electrodes mounted on Huxley–Wall or motorized micromanipulators (Sutter Instruments, Novato, CA, USA). To target cones, we crossed HRGP-Cre mice with Ai14 mice that express Td-Tomato under control of the Cre recombinase. Cones can also be distinguished from rods by their larger membrane capacitance and larger Ca^2+^ currents (I_Ca_).

Rod ribbons are surrounded by glutamate transporters EAAT2 and EAAT5 [52,53,54,55], and glutamate binding to these transporters activates a large anion conductance [56,57,58,59]. While glutamate transporter anion currents (I_A(glu)_) are activated by glutamate binding and uptake, they are thermodynamically uncoupled from the transport process [37,60]. I_A(glu)_ was enhanced by replacing Cl^−^ with thiocyanate (SCN^−^) in the patch pipette [56,61]. The intracellular pipette solution for these experiments contained 120 mM KSCN, 10 mM TEA-Cl, 10 mM HEPES, 1 mM CaCl_2_, 1 mM MgCl_2_, 0.5 mM Na-GTP, 5 mM Mg-ATP, 5 mM EGTA, 5 mM phosphocreatine, pH 7.3.

Whole-cell recordings were performed using an Axopatch 200B amplifier (Molecular Devices, San Jose, CA, USA), and signals were digitized with DigiData 1550 (Molecular Devices). Data acquisition and analysis were performed using pClamp 10 (Molecular Devices) or Axo-Graph software. Currents were filtered at 1–2 kHz. Recordings of I_A(glu)_ and I_Ca_ were leak-corrected using P/6 subtraction. Voltages were not corrected for liquid junction potentials (KSCN pipette solution: 3.9 mV). Chemical reagents were obtained from Sigma-Aldrich unless otherwise indicated.

### 4.5. Immunohistochemistry

For immunohistochemical experiments, eyes were enucleated after euthanizing the mice and placed in oxygenated Ames’ medium. After removing the cornea and the lens, the posterior eyecup was fixed in 4% paraformaldehyde for 45 min. The eyecup was then washed in PBS three times for 10 min each and cryoprotected in 30% sucrose overnight at 4 °C. The eyecups were embedded in an OCT compound (Sakura Finetek USA, Torrance, CA, USA) and stored at −80 °C until sectioning at 25 µm with a cryostat (Leica CM 1800, Chicago, IL, USA). Retinal sections were treated with a blocking solution of 1% Triton X-100 and 6% donkey serum (Jackson Labs, Bar Harbor, ME, USA) for 2 h before applying the primary antibody. Primary and secondary antibodies (Table 1) were diluted to working concentrations in blocking solution. The sections were incubated in the primary antibody at 4 °C overnight and in the secondary antibody at room temperature for 2–3 h. The retinal sections were mounted with Vectashield (Vector Labs, Burlingame, CA, USA, RRID: AB_2336787) before imaging.

Confocal imaging was performed using NIS Elements software (Nikon Instruments, Melville, NY, USA, RRID: SCR_014329) and a spinning disk confocal microscope that consisted of a laser confocal scan head (Perkin Elmer Ultraview LCI, Waltham, MA, USA) equipped with a cooled CCD camera (Hamamatsu Orca ER; Hamamatsu City, Shizuoka, Japan, RRID: SCR_017105) mounted on a Nikon E600FN microscope. Fluorescent excitation was delivered from an argon/krypton laser at 488, 568, or 648 nm, and emission was collected at 525, 607, and 700 nm, respectively. The filters were controlled using a Sutter Lambda 10–2 filter wheel and a controller. The objective (water immersion, 60×, 1.2 NA) was controlled with an E662 Z-axis controller (Physik Instrumente; Karlruhe, Germany). The images were analyzed and adjusted using Nikon Elements, Fiji, and Adobe Photoshop software.

### 4.6. Electron Microscopy

The retinal pieces were fixed overnight at 4 °C in 2% glutaraldehyde, 2% paraformaldehyde, and 0.1 M Sorensen’s phosphate buffer (pH 7.4). After fixation, the retinas were washed three times in phosphate-buffered saline and then placed in 1% osmium tetroxide. The samples were dehydrated through a graded ethanol series with each concentration (50%, 70%, 90%, 95%, 100%) applied for 3 min. The retinas were then washed three times with 100% propylene oxide. The samples were left overnight in a 1:1 mixture of the Araldite embedding medium and propylene oxide, embedded in fresh Araldite in silicon rubber molds, and then placed in an oven at 65 °C overnight. The resulting blocks were thin-sectioned on a Leica UC6 ultramicrotome and placed on 200 mesh copper grids. The sections were stained with 1% uranyl acetate and Reynold’s lead citrate. The sections were examined in a FEI Tecnai G2 TEM operated at 80 kV.

### 4.7. Experimental Design and Statistical Analysis

Statistical analysis and data visualization were performed using Clampfit 10 (Molecular Devices, San Jose, CA, USA) and GraphPad Prism 9 software (GraphPad Software, San Diego, CA, USA. Roughly equal numbers of male and female mice were used for these experiments. For ERG measurements, we analyzed the sample for each condition using Dunnett’s multiple-comparisons test with one-way ANOVA. When comparing different experimental conditions using I_A(glu)_ or horizontal cell synaptic currents, we used multiple *t*-tests with Holm–Sidak correction and one-way ANOVA. The criterion for statistical significance was set at α = 0.05. Error bars in the figures show 95% confidence intervals. The numerical data are reported as the means ± SD unless otherwise indicated.

## Figures and Tables

**Figure 1 ijms-23-06429-f001:**
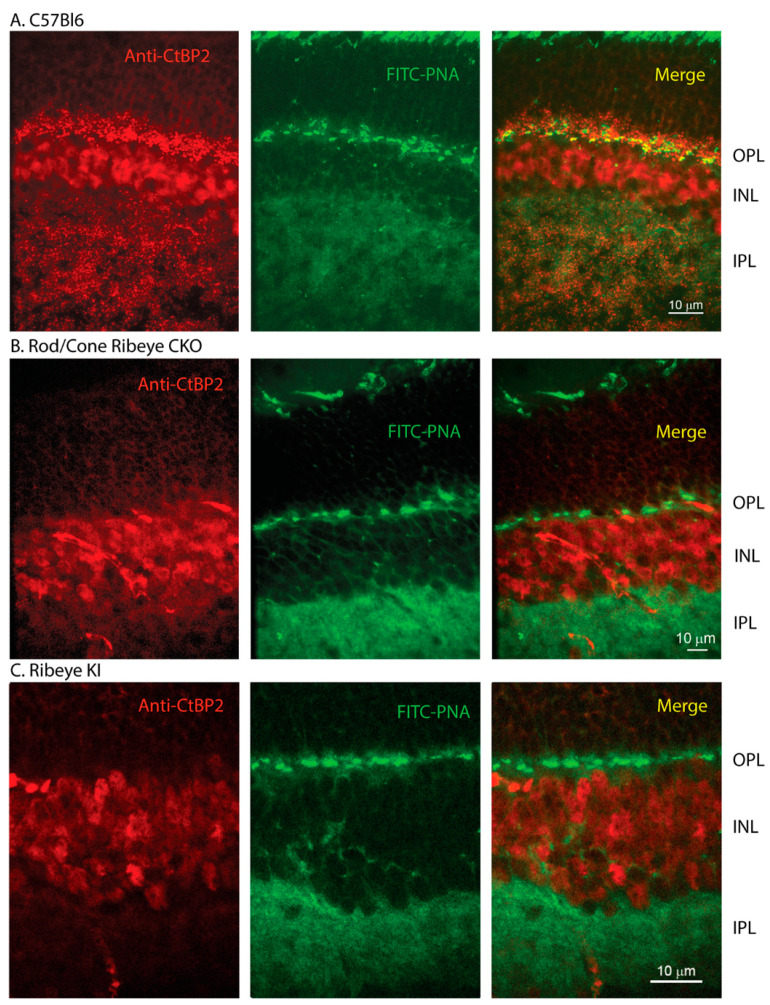
Synaptic ribbons are absent from retinas of both rod/cone *Ribeye* CKO and *Ribeye* KI mice. (**A**) Wild-type C57Bl6J retina labeled with a primary antibody to the B-domain of RIBEYE (Synaptic Systems, 192-003) and visualized with rhodamine-conjugated secondary antibody-labeled ribbons in the OPL and IPL along with somas in the INL. The base of cone pedicles was strongly labeled by FITC-conjugated peanut agglutinin (FITC-PNA, green). (**B**) Ribbon labeling with the B-domain antibody was eliminated from the OPL, but also from the IPL in rod/cone *Ribeye* CKO retina (Rho-iCre × HRGP-Cre × *Ribeye*^fl/fl^). Labeling of the INL remained in place. (**C**) Ribbon labeling was also eliminated from retinas of *Ribeye* KI mice while labeling of the INL remained.

**Figure 2 ijms-23-06429-f002:**
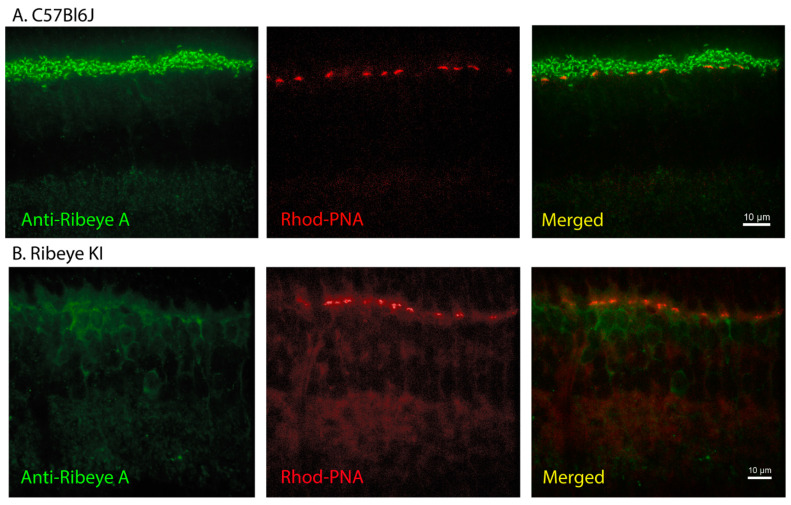
Labeling of synaptic ribbons with the A-domain antibody was eliminated in the *Ribeye* KI mice. (**A**) Wild-type C57Bl6J retina labeled with the primary antibody to the A-domain of RIBEYE (Synaptic Systems, 192-104) and visualized with an Alex488-conjugated second antibody strongly labeled ribbons in the OPL with weaker labeling of the IPL. Cone pedicles were labeled with rhodamine-conjugated PNA (red). (**B**) Labeling of ribbons in the OPL and IPL was absent from the *Ribeye* KI retinas. Weak labeling was seen in the outer region of the INL where bipolar cell bodies reside, suggesting that replacing the B-domain with GCamP3 impaired RIBEYE protein processing.

**Figure 3 ijms-23-06429-f003:**
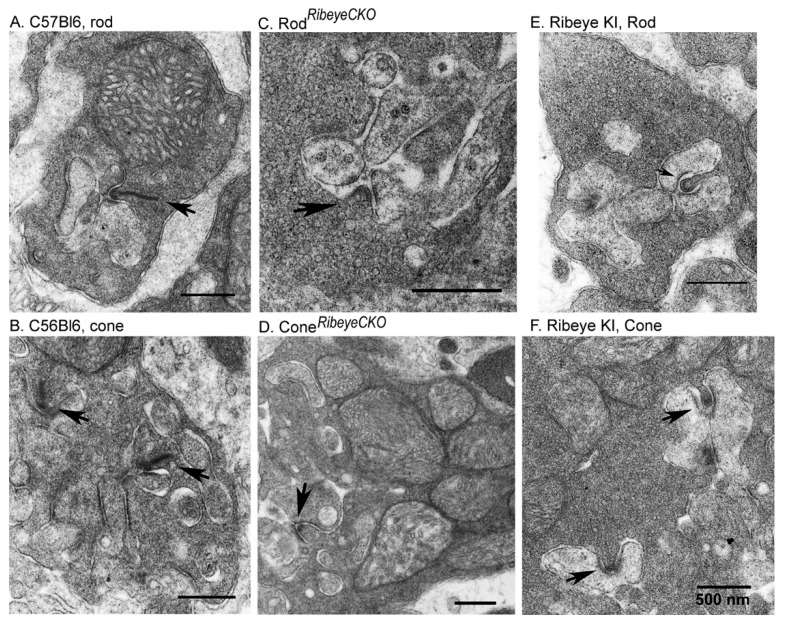
Electron micrographs of rod and cone terminals show that ribbons were eliminated from the conditional *Ribeye* knockouts and *Ribeye* KI retinas. (**A**) Representative image of a rod terminal from a wild-type C57Bl6J mouse retina. The arrow points out the location of the ribbon. (**B**) Representative image of a cone terminal. (**C**) Image of a rod terminal from a rod*^Ribeye^*^CKO^ mouse. An arciform density (arrow) remains but the associated ribbon is absent. (**D**) Image of a cone terminal from a cone*^Ribeye^*^CKO^ retina showing an arciform density with no associated ribbon. (**E**,**F**) Similar examples from a rod and a cone, respectively, from a *Ribeye* KI retina.

**Figure 4 ijms-23-06429-f004:**
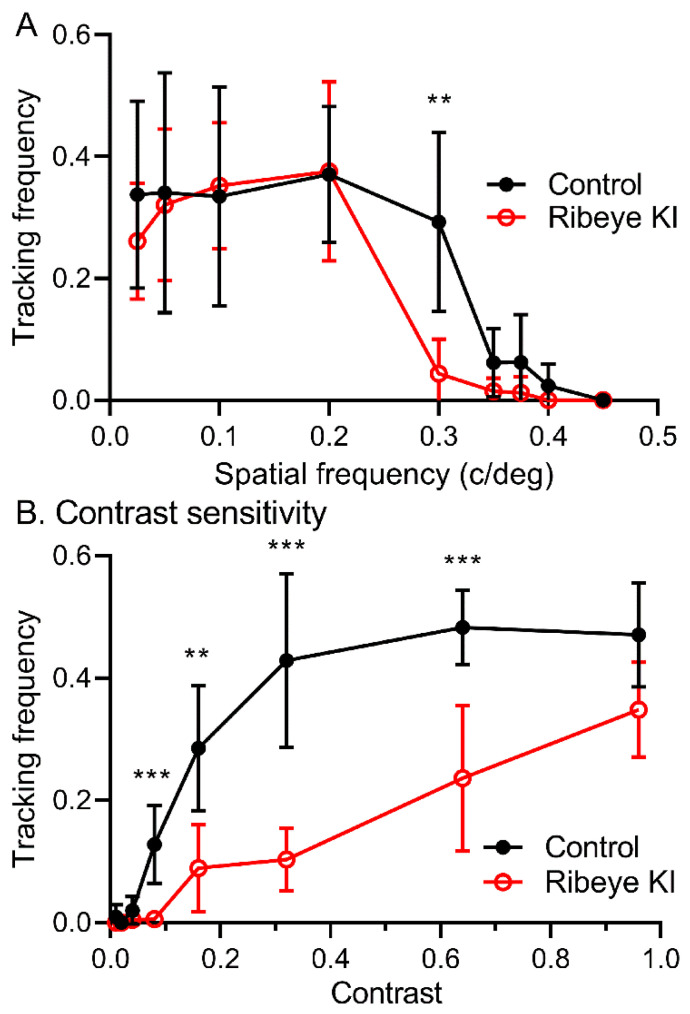
Optomotor responses showed diminished responses in the *Ribeye* KI mice lacking ribbons. (**A**) Frequency of the trials in which mice tracked a rotating stimulus as a function of spatial frequency using 100% contrast gratings at nine spatial frequencies (cycles/degree): 0.025, 0.05, 0.10, 0.20, 0.30, 0.35, 0.375, 0.40, and 0.45. Acuity was diminished significantly in the *Ribeye* KI mice (*n* = 14) compared to the control C57Bl6J mice (*n* = 12) at 0.3 cycles/deg (**, *p* = 0.00116, multiple *t*-tests corrected for multiple comparisons using the false discovery rate approach). (**B**) Tracking frequency as a function of contrast. Contrast sensitivity was assessed by presenting 0.20 cycles/degree gratings with the following luminance contrasts (%): 0.01, 0.02, 0.04, 0.08, 0.16, 0.32, 0.64, and 0.96. Sensitivity was diminished significantly in the *Ribeye* KI mice (*n* = 14) compared to the control C57Bl6J mice (*n* = 12) at 0.08 (***, *p* = 0.00017), 0.16 (**, 0.00179), 0.32 (***, *p* = 0.00004), and 0.64% contrast (***, *p* = 0.00089; multiple *t*-tests, false discovery rate approach). Stimulus velocity was fixed at 12 deg/s for all procedures. The error bars in this and all subsequent figures show 95% confidence intervals.

**Figure 5 ijms-23-06429-f005:**
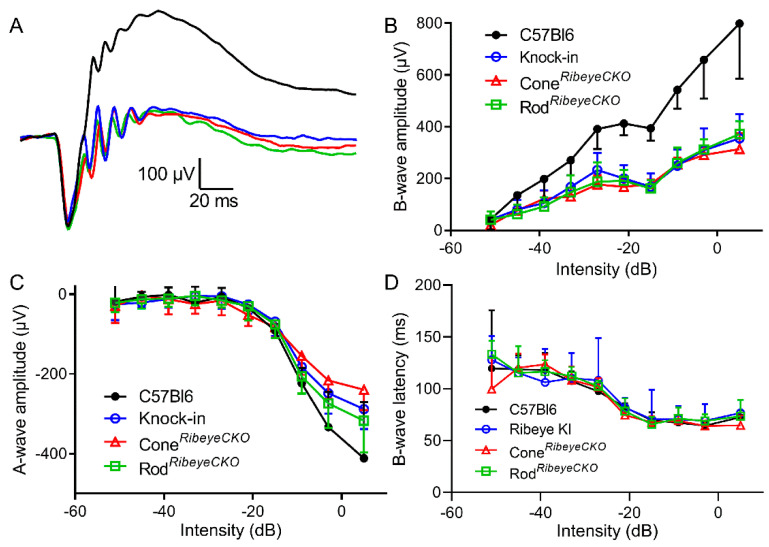
Scotopic ERG b-waves were reduced equally in the cone*^Ribeye^*^CKO^, rod*^Ribeye^*^CKO^, and *Ribeye* KI retinas. (**A**). Example waveforms. (**B**). Plot of the b-wave amplitude as a function of flash intensity measured in the C57Bl6J (*n* = 4 mice), cone*^Ribeye^*^CKO^ (*n* = 5), rod*^Ribeye^*^CKO^ (*n* = 5), and *Ribeye* KI retinas (*n* = 5). (**C**). A-wave amplitude as a function of intensity. (**D**). B-wave latency vs. intensity. The error bars show the SD.

**Figure 6 ijms-23-06429-f006:**
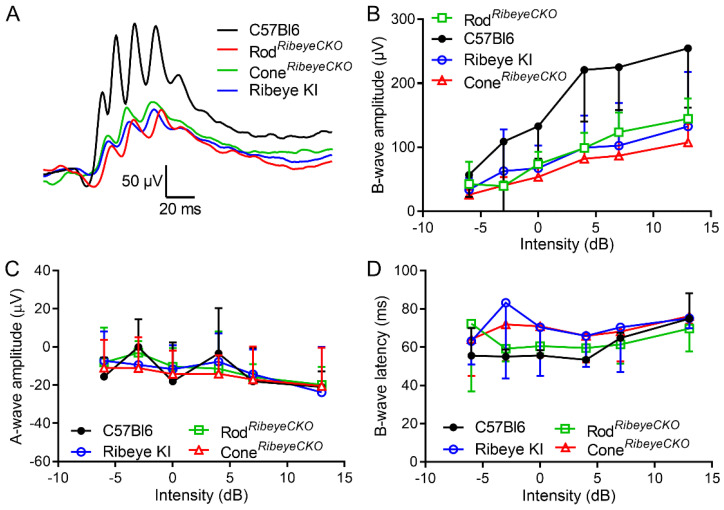
Photopic ERG b-waves were reduced equally in the cone*^Ribeye^*^CKO^, rod*^Ribeye^*^CKO^, and *Ribeye* KI retinas. (**A**) Example waveforms. (**B**) B-wave amplitude as a function of flash intensity measured in the C57Bl6J (*n* = 5 mice), cone*^Ribeye^*^CKO^ (*n* = 5), rod*^Ribeye^*^CKO^ (*n* = 5), and *Ribeye* KI retinas (*n* = 4). (**C**) A-wave amplitude as a function of intensity. (**D**) B-wave latency vs. intensity.

**Figure 7 ijms-23-06429-f007:**
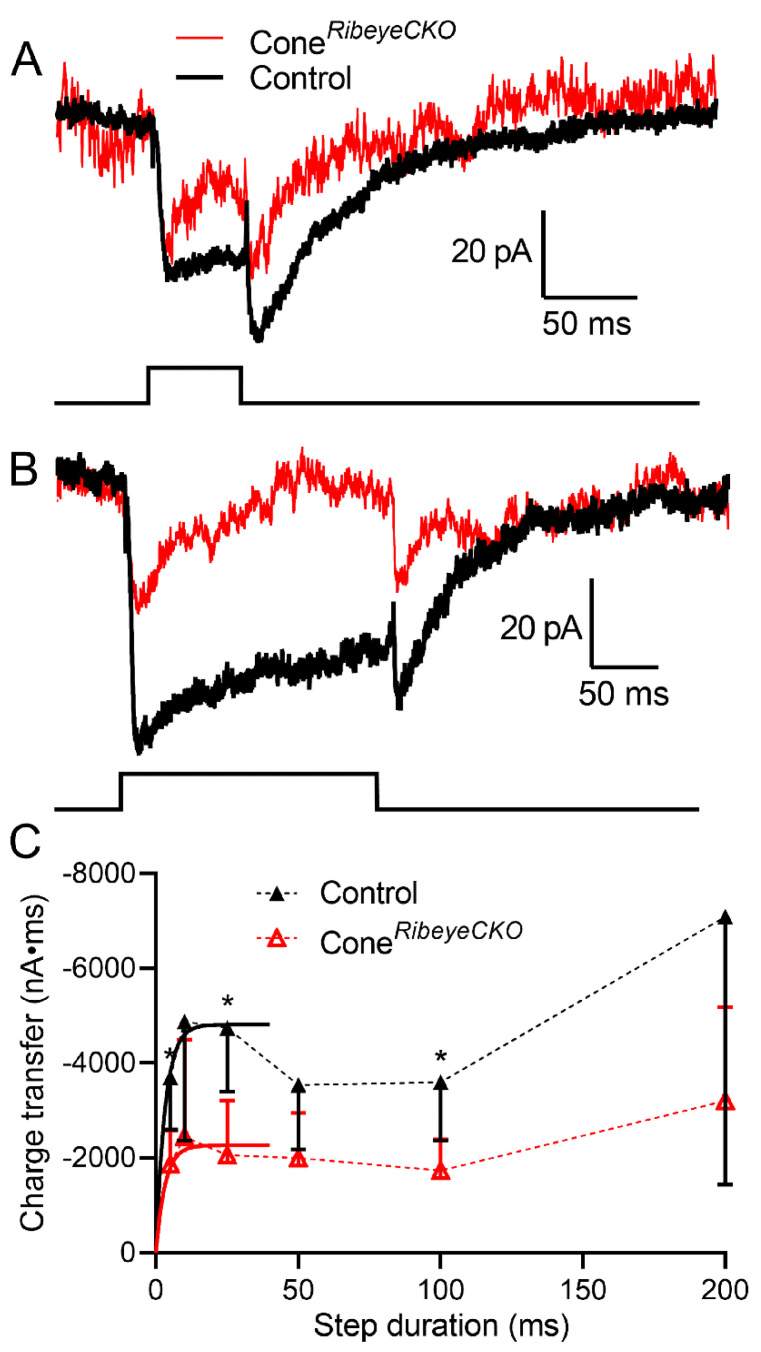
Eliminating cone ribbons halved the releasable vesicle pool. Example recordings of I_A(glu)_ evoked by steps of 50 and 200 ms recorded from a cone in a control C57Bl6J retina ((**A**), black traces) and a cone in a cone*^Ribeye^*^CKO^ retina ((**B**), red traces). (**C**). Plot of the I_A(glu)_ charge transfer measured following steps of 5, 10, 25, 50, 100, and 200 ms from cones in the C57Bl6J (*n* = 14–28 cells at each data point) and cone*^Ribeye^*^CKO^ (*n* = 5–18 cells) retinas. The data from 5–25 ms were fit with single exponentials yielding the best-fit amplitude of 4835 and 2182 pA*ms in the wild-type and *Ribeye* CKO cones, respectively. The best-fit time constants were 3.32 and 2.43 ms in the wild-type and *Ribeye* CKO cones, respectively. These did not differ significantly (*p* = 0.83, F-test). The overall sample of responses to different test durations was significantly smaller in the cones lacking RIBEYE (*p* < 0.0001, ANOVA). The differences were also significantly different at individual timepoints of 5, 25, and 100 ms (*: *p*<0.05; adjusted *p*-values, Holm–Sidak method for multiple comparisons, *p* = 0.0292, 0.0176, and 0.0347, respectively).

**Figure 8 ijms-23-06429-f008:**
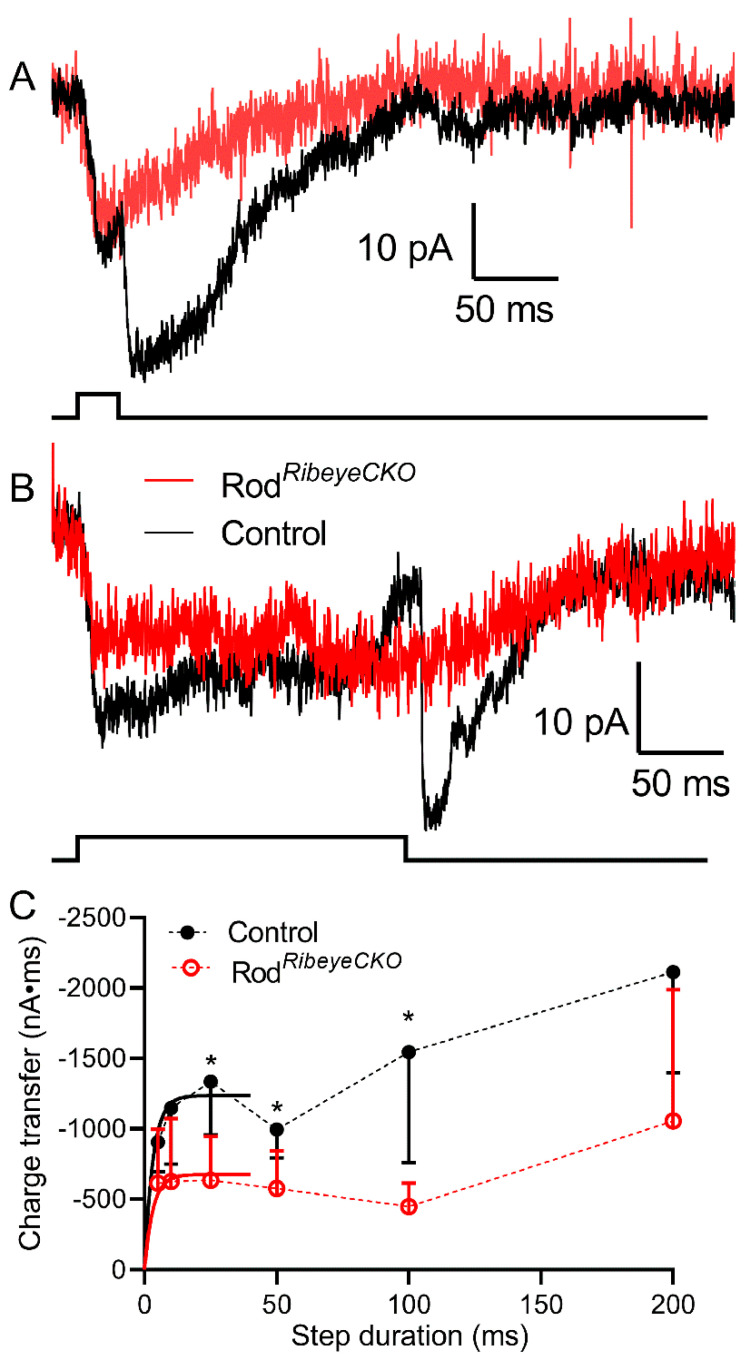
Eliminating rod ribbons halved the releasable vesicle pool. Example recordings of I_A(glu)_ evoked by steps of 10 and 200 ms recorded from a rod in control C57Bl6J ((**A**), black traces) and a rod in rod*^Ribeye^*^CKO^ retina ((**B**), red traces). (**C**). Plot of I_A(glu)_ charge transfer measured the following steps of 5, 10, 25, 50, 100 and 200 ms from rods in the C57Bl6J (*n* = 13–34 cells at each data point) and rod*^Ribeye^*^CKO^ (*n* = 9–18 cells) retinas. The data from 5–25 ms were fit with single exponentials yielding the best-fit amplitude of 1326 and 632 pA*ms in the wild-type and *Ribeye* CKO rods, respectively. The best-fit time constants were 4.47 and 1.40 ms in the wild-type and *Ribeye* CKO rods, respectively. These did not differ significantly (*p* = 0.44, F-test). Differences between the rods with and without ribbons were significantly different at 25, 50, and 100 ms (*: *p*<0.05: adjusted *p*-values, Holm-Sidak method for multiple comparisons, *p* = 0.0277, 0.0484, and 0.0424, respectively).

**Figure 9 ijms-23-06429-f009:**
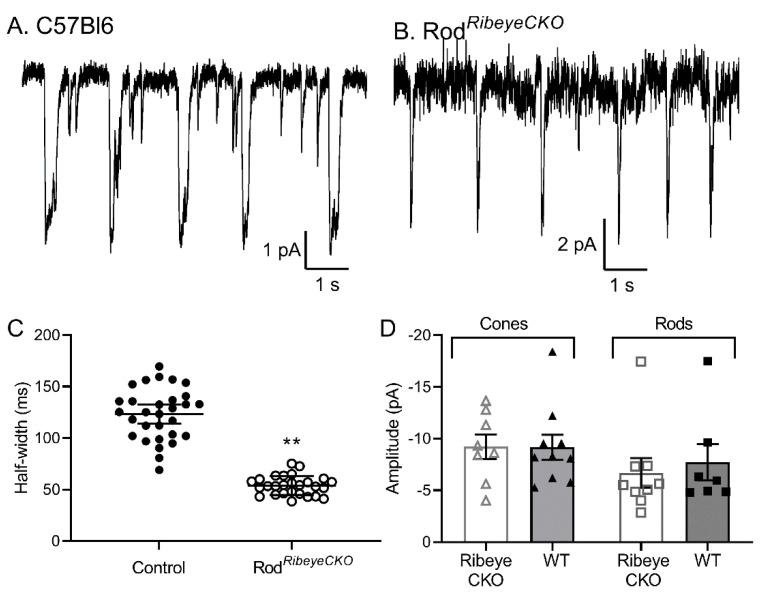
The duration of a periodic multiquantal release event was halved in the rod*^Ribeye^*^CKO^ mice. (**A**,**B**) Examples illustrating the appearance and regularity of multiquantal release events in the mice with (**A**) and without (**B**) ribbons. (**C**) Scatter dot plot along with the means and SD of the half-widths measured from multiquantal bursts recorded at a holding potential of −40 mV in the control and rod*^Ribeye^*^CKO^ rods (132 ± 64 ms, *n* = 10 control rods, 167 events; 55 ± 26 ms, *n* = 5 rod*^Ribeye^*^CKO^ rods, 98 events; **: *p* = 0.003, nested *t*-test). (**D**). Average amplitude of spontaneous I_A(glu)_ events recorded from cones in the cone*^Ribeye^*^CKO^ (9.22 ± 3.33 pA, *n* = 8 cones) and wild-type (9.16 ± 3.39 pA, *n* = 10) retinas and rods in the rod*^Ribeye^*^CKO^ (6.68 ± 4.29 pA, *n* = 9 rods) and wild-type (7.71 ± 4.63 pA, *n* = 7) retinas.

**Figure 10 ijms-23-06429-f010:**
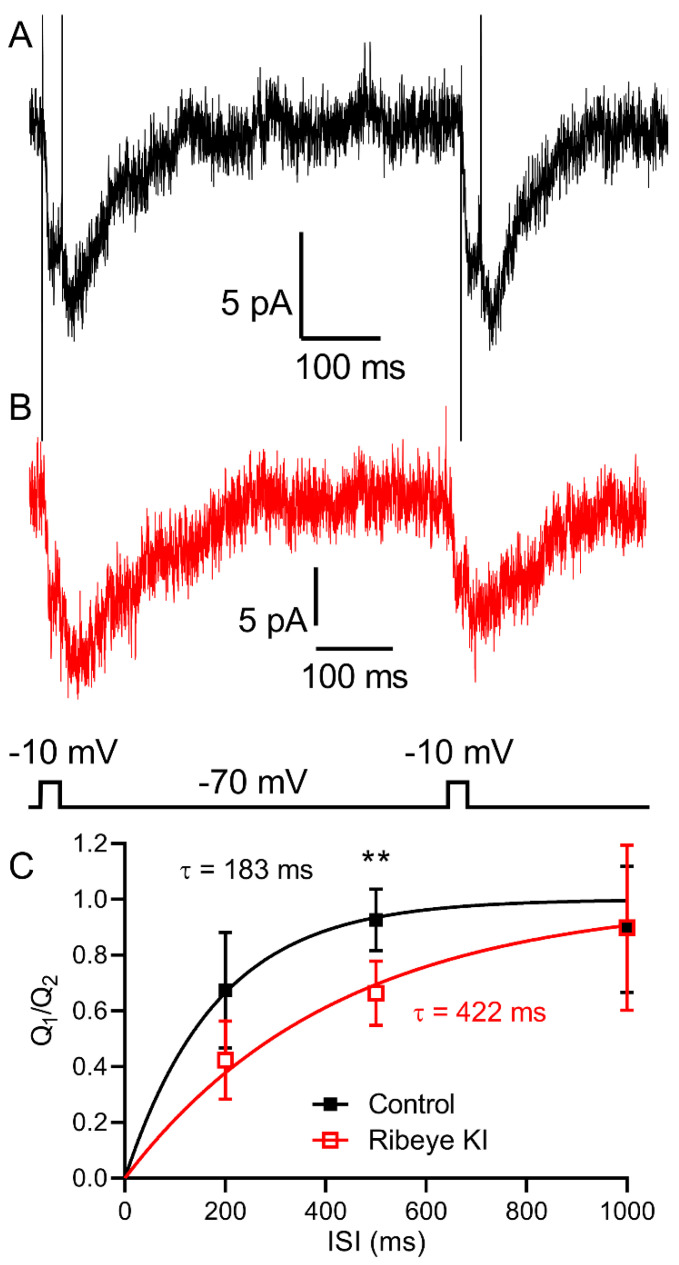
Loss of ribbons slowed down recovery from paired pulse depression in the rods assessed by measuring I_A(glu)_. Examples of I_A(glu)_ recorded from a control C57Bl6 rod ((**A**), black trace) and a *Ribeye* KI rod ((**B**), red trace) that lack ribbons. The test steps in these examples were separated by 500 ms. (**C**). Amplitude of the I_A(glu)_ charge transfer evoked in a rod by the second test step relative to the first response in each trial from both control (black squares, *n* = 8–10 rods at each datapoint) and *Ribeye* KI rods (open red squares, *n* = 9–12 rods from five mice). Individual values differed significantly at the 500 ms interval (**: *p* = 0.002, *t*-tests corrected for multiple comparisons). Recovery of the I_A(glu)_ charge transfer from paired pulse depression was fit with a single exponential function. In the absence of ribbons, the time constant doubled (*p* = 0.026; extra sum-of-squares F-test) from 183 ms in the C56Bl6 rods (*n* = 10 rods; rate constant, K = 0.00547 ± 0.00102 ms, SEM) to 422 ms in the *Ribeye* KI rods (*n* = 12; K = 0.002372 ± 0.000351 ms).

**Figure 11 ijms-23-06429-f011:**
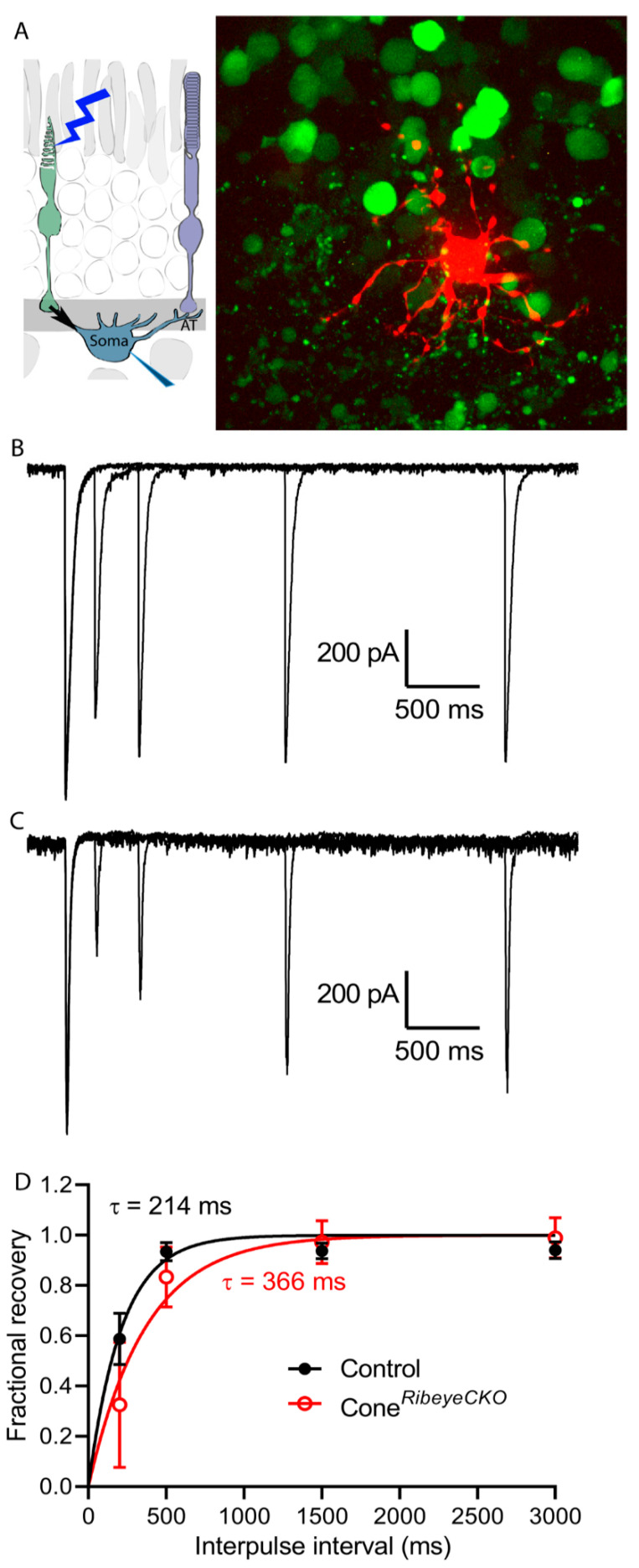
Loss of ribbons slowed down recovery from paired pulse depression in cones assessed by optogenetic stimulation of postsynaptic currents in horizontal cells. (**A**) Diagram illustrating optogenetic stimulation of cones while recording from a horizontal cell. The photomicrograph shows a horizontal cell filled with sulforhodamine B through a patch pipette (red) within horizontal retinal slice preparation. The retina was also loaded with a Ca^2+^-sensitive dye, Cal520 (green), revealing some underlying cones. (**B**) Example horizontal cell current recordings showing responses to a series of paired pulse trials obtained with optogenetic flashes applied at different inter-pulse intervals (200, 500, 1500, 3000 s). The responses in (**B**) are from a horizontal cell in C57Bl6J control mouse retina while (**C**) illustrates paired pulse trials from a cone*^Ribeye^*^CKO^ horizontal cell. (**D**) Plots amplitude of the horizontal synaptic current evoked by the second flash relative to the current amplitude evoked by the first flash in each trial. Responses recovered with a time constant of 214 ms (K = 0.004665 ± 0.0003175) in the control retinas (*n* = 11 horizontal cells in four mice), but more slowly in the cone*^Ribeye^*^CKO^ retinas (*n* = 10 horizontal cells in four mice) with a time constant of 366 ms (K = 0.002732 ± 0.0003716 ms; *p* = 0.0002, F-test).

**Figure 12 ijms-23-06429-f012:**
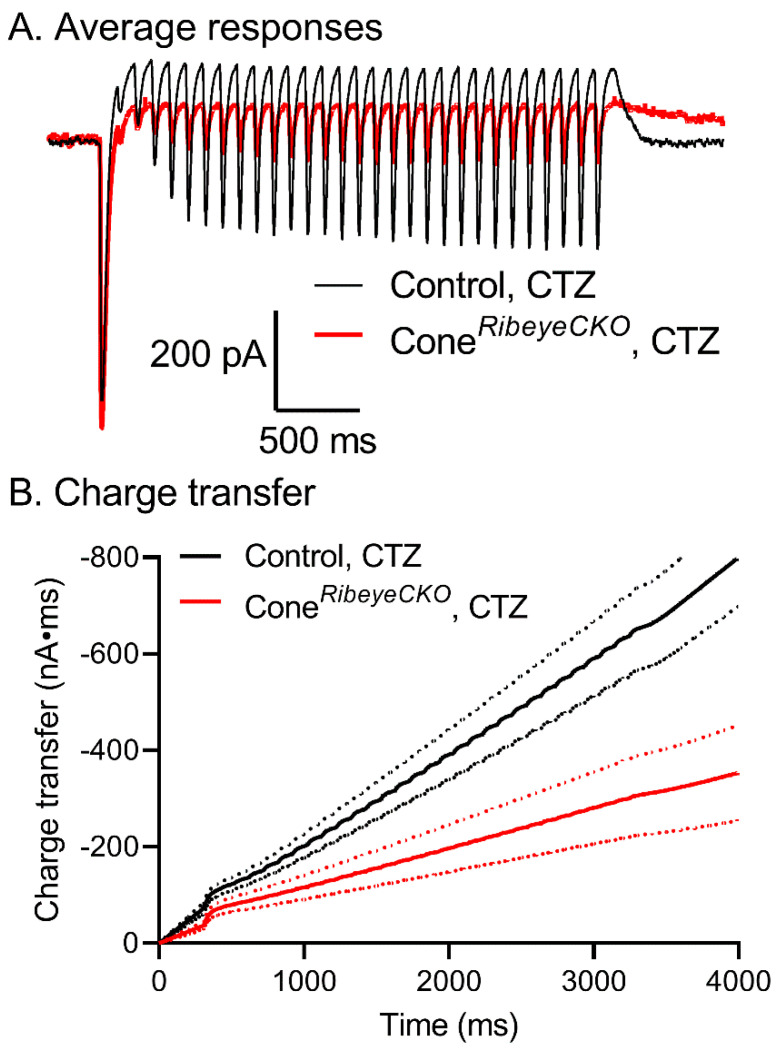
Pulse train optogenetic stimuli show impaired replenishment at cone synapses of the mice lacking ribbons. (**A**) Average recordings of horizontal cell currents evoked by a train of optogenetic pulses. Black trace: control C57Bl6J mice (*n* = 11 horizontal cells in four mice). Red trace: cone*^Ribeye^*^CKO^ mice (*n* = 10 horizontal cells in four mice). (**B**) Mean ± 95% confidence interval of the cumulative increase in the synaptic charge transfer during pulse trains in the same sample of the control and cone*^Ribeye^*^CKO^ horizontal cells. The charge transfer rose more slowly in horizontal cells of the cone*^Ribeye^*^CKO^ retinas vs. the control retinas, consistent with slower replenishment without ribbons.

**Table 1 ijms-23-06429-t001:** Reagents.

Reagent Type	Designation	Source or Reference	Identifiers	Additional Information
Antibody	anti-Ctbp2	BD Biosciences	Cat No. 612044	Mouse (1:1000)
Antibody	Ribeye-A	Synaptic Systems	Cat No. 192-103	Rabbit (1:200)
Antibody	Goat anti-Rb	Life Technologies	REF No. A11034	Alexa 488 (1:200)
Antibody	Goat anti-Ms	Invitrogen	REF No. 6393	Rhodamine (1:200)
PNA	Rhodamine	Vector	REF No. RL-1072	1:50
PNA	FITC	BIONEXUS	LOT No. 23513	1:50

## Data Availability

The datasets generated during the current study are available from the corresponding author on request.

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
