# Peer review of "Eliminating Synaptic Ribbons from Rods and Cones Halves the Releasable Vesicle Pool and Slows Down Replenishment"

_ijms, 2022, doi:10.3390/ijms23126429_

Round 1

Reviewer 1 Report

There is no major objection, it is an excellent piece of work.

Figures 5 and 6, legends. I do not see comments for Figures 5D and 6D in the legends, 2nd C may rather be D.

Figures 4, 5, 6, 7C, 10C, 11D. Present error bars only in one direction, either above or below, and in opposite direction in case there are two or more curves. This would make figures less crowded.

Author Response

Thanks for the kind comments and catching errors in the figure legends.  We  corrected legends of Figs. 5 and 6 and modified the ERG figures comparing multiple genotypes to show error bars in a single direction as recommended.  For the figures in which we compared only two genotypes, we felt there was value in showing both directions for error bars.

Reviewer 2 Report

In this manuscript the Authors examine glutamate release from rod and cone photoreceptors upon manipulating or eliminating the major component of presynaptic ribbons, the RIBEYE protein. Multiple transgenic mouse lines are used and compared for their presynaptic function: the Ribeye KI, with modified Ribeye and conditional, cell specific ribeye KOs in which rods or cones are lacking ribeye. The Authors first assess these transgenic lines with immunohistochemistry as well as electron microscopy for the presence of ribbons in photoreceptors then upon validation (i.e. proving that ribbons are missing) the characteristics of glutamate release from these ribbon-less photoreceptors are studied, as well as the consequences of altered synaptic function at cellular, circuit, and visual behavior level.

 The results are consistent with previous reports in that disruption of ribbons seriously impacts the synapses of rods and cones, reduces the readily releasable pool of synaptic vesicle, slows their replenishment that presumably contributes to reduced b-waves in ERG recordings as well as deficient optomotor responses.   

Overall, the manuscript reports on a nice set of experiments that are consistent with and support each other. The manuscript is well organized, clearly written and illustrated.  The statistical analysis is appropriate.  The interpretations are supported by the data. 

I have a few minor comments: 

1.       The Authors use C57Bl6 mice as a control, but no mention on the background of the transgenic lines, or if backcrossing onto C57Bl6 for several generations took place that is a prerequisite if the line was generated on different background. Even if the mouse line details are available on   the Jax site, a related clarifying statement in the Mice section is necessary. 

2.       Ln 204-205: ”Passive membrane resistance was subtracted from…..” as well as ln 417-418:”Passive membrane properties were subtracted using P/6 protocols” I assume in both cases leak was corrected,  using a P/6 protocol. Consider stating it accordingly.

3.       Detecting reduced synaptic function upon eliminating the ribbon from a ribbon synapse is somewhat expected. The discussion mainly reiterates the results. I personally found it more intriguing, that substantial glutamate release still takes place from photoreceptors with no ribbon. I would rather see some discussion/speculation on how important the ribbons might be after all, based on these results?

Author Response

Thank your for your kind comments and helpful suggestions.  We added information on the backgrounds of mice used in this study (lines 96-98) and corrected wording regarding P/6 subtraction (lines 207-208).

We were also surprised that eliminating ribbons has such modest effects.  We added a short discussion of this at the end of the paper, along with citation of a modeling study suggesting that vesicles can be rapidly delivered to rod bipolar cell release sites even without ribbons (lines 685-690).